# The Effect of Lure Position and Vegetation on the Performance of YATLORf Traps in the Monitoring of Click Beetles (*Agriotes* spp., Coleoptera: Elateridae)

**DOI:** 10.3390/insects14060542

**Published:** 2023-06-10

**Authors:** Lorenzo Furlan, Stefano Bona, Miklós Tóth

**Affiliations:** 1Veneto Agricoltura, Agricultural Research Department, Viale dell’Università, 14, 35020 Legnaro, Italy; 2Department of Agronomy, Food, Natural Resources, Animals and Environment, University of Padova, 35020 Legnaro, Italy; stefano.bona@unipd.it; 3Centre for Agricultural Research, Plant Protection Institute, ELKH, H-1022 Budapest, Hungary; toth.miklos@atk.hu

**Keywords:** IPM, *Agriotes brevis*, *A. ustulatus*, *A. sordidus*, *A. lineatus*, *A. litigiosus*

## Abstract

**Simple Summary:**

The EU has established ambitious goals for pesticide reduction. The achievement of these goals requires the introduction of easy-to-use, low-cost and effective tools to monitor crop pests. Sex pheromone YATLORf (Yf) traps have proven to be a reliable low-cost tool for monitoring *Agriotes* spp., which are the most harmful group of soil pests in Europe. They can damage a wide range of arable crops, including potato, maize and sunflower. In order to optimize Yf use, this study explored the effect of lure position in the trap and vegetation field cover density on trap performance. Lure attractant capacity varied greatly with the lure’s position in the trap and the extent of the vegetation density around the trap. Guidelines for making practical decisions are given. The ‘low’ lure position in the Yf is suitable for all species in all field conditions and is the best choice for *A. brevis*; the ‘medium’ position should be considered for *A. brevis* only when vegetation is dense. The ‘high’ lure position is unsuitable for *A. brevis* and *A. obscurus*, and should be considered for some species only. Dense vegetation (e.g., winter wheat) reduced the potential for catching *A. sordidus*. Placing the trap just outside the field, or in a nearby field with bare/sparse vegetation, maintained the maximum catching potential. Vegetation density was found to influence beetle sex ratio, with no or very few female *A. brevis* and *A. sordidus* being found in traps placed in the middle of fields with dense vegetation. This research has made it possible to conduct studies on multi-baiting the same trap, which can significantly reduce monitoring costs.

**Abstract:**

Low-cost monitoring tools are needed to implement IPM in arable crops. YATLORf (Yf) traps baited with respective synthetic pheromone lures have proven to be a reliable tool for monitoring *Agriotes* spp., Europe’s most harmful soil pests. To optimize Yf use, we studied the effect of lure position in the trap and crop density on trap performance. Yf management detail was studied between 2000–2003 and 2014–2016 in various countries, with the traps being arranged in blocks. Each block contained one trap per treatment (i.e., lure position) under study. It was ascertained that lure attractant capacity can vary greatly with the lure’s position in the trap and the extent of vegetation. Information for making practical decisions is given. The ‘low’ lure position is suitable for all species in all field conditions, and is the best choice for *A. brevis*. Lures for *A. brevis and A. lineatus* need to be placed in the low position when the field has no or sparse vegetation cover. The ‘high’ lure position is unsuitable for *A. brevis* and *A. obscurus*, and should be considered for some species only. There are no restrictions on position for catching *A. sordidus*, i.e., any position is suitable. Dense vegetation (e.g., wheat) reduced the Yf trap’s potential for catching *A. sordidus*. Placing the trap just outside the field, or in a nearby field with bare/sparse vegetation, maintained the maximum catching potential. Vegetation density also influenced beetle sex ratio, with *A. brevis* and *A. sordidus* females always found in traps placed in fields with bare or low-density vegetation. Our findings have made it possible to obtain consistent monitoring outputs and to begin studies on multi-baiting the same trap, which can significantly reduce monitoring costs.

## 1. Introduction

European and non-European farmers, particularly in North America, commonly apply a soil insecticide (mainly neonicotinoids and pyrethroids) to maize and a range of other arable crops at planting [1]. In the cases of maize and winter wheat, two of the world’s main arable crops, the area of cultivated land treated with soil insecticides to control wireworms and other soil pests vastly exceeds the area with a tangible risk of economic damage [1], leading to unjustified environmental impact. The common prophylactic use of pesticides, which has a major impact on biodiversity, ecosystems and ecosystem services worldwide [2], is pointless due to the low number of fields with populations exceeding thresholds [3,4], and goes against the commonly recognized principles of Integrated Pest Management (IPM).

Therefore, since arable farming often has limited resources in terms of income, labor and technology, finding low-cost reliable monitoring tools is vital for IPM implementation in arable crops. Tools such as pheromone traps can be very important for affordable risk assessment of soil pest damage so that farmers and IPM advisors can apply control solutions, including insecticides, only where they are needed, thus ensuring that the objectives of Directive 128/2009/EC of the European Parliament are met.

A tremendous amount of knowledge has been collected on the pheromones of moths (Lepidoptera) since a female moth-produced pheromone was first identified chemically [5,6,7]. As a result of intensive research efforts over the past few decades, pheromone lures have also been developed and optimized for all important click beetles of the *Agriotes* genus in Europe [8,9]. Only over the last few years, however, have suitable sex pheromone traps for monitoring the most important Coleoptera soil pests (click beetles, fam. Elateridae) become available [3,6]. Known as YATLORf (Yf) traps, they have proven to be an effective, low-cost tool for monitoring adults of *Agriotes* spp., Europe’s most important soil pests, as well as being essential for IPM of the main arable crops [3].

In order to make Yf catches consistent and suitable for threshold set-up, some key trap management solutions need to be established, as we believed that the lure position in the trap and the Yf position in the field may impact catch potential. Consequently, this present study was designed to determine:(a)which lure positions in the traps were most effective at catching a target species;(b)the effect of vegetation/crop density and trap location within the field on trap effectiveness;(c)the effect of vegetation presence and density on the number of males and females.

Herein, we summarize the above experiments, which cover two distinct time periods: 2000–2003 and 2014–2016. For further details, see the following section.

## 2. Materials and Methods

YATLORf (Yf) traps [10] (Figure 1) are widely used for the pheromone trapping of adult click beetles in Europe [11], with our field trials using Yf traps [3,11] produced by ROSA Micro S.r.l. Ceggia, Venice, Veneto, Italy. The pheromone lures used were part of the CSALOMON^®^ trap family (Plant Protection Institute, CAR, Budapest, Hungary). Compositions of single lures were based on earlier reports in the literature [8,12,13,14] (Table 1).

Field trapping tests were conducted at a number of sites in Italy, Portugal and Slovenia from 2001 to 2003 (Table 2) using accepted methods for trapping experiments [5]. The Yf trap’s white bottom was placed face-down, with its brown edge 1–2 cm beneath the soil. The baits were managed as per a standard seasonal schedule based upon the life cycle and behavior of each species. For example, in northeast Italy *A. ustulatus* click beetles swarm from early June to August, and *A. sordidus* beetles swarm from April to August, peaking in May; much longer than *A. ustulatus*. *A. brevis* behaves in a similar manner to *A. sordidus*, but has a longer swarming period, which starts slightly earlier than that of *A. sordidus* [11].

*A. brevis* lures were not replaced, but the other lures were replaced every 30 to 40 days when the experiments ran for more than one month.

On every inspection, the bottom was cleaned up, with soil and any residue being removed. The insects were removed from the trap with the following procedure: the trap was unearthed; it was then placed inside a large plastic bag, which was opened so that the insects dropped inside; the bag was closed immediately after the trap had been removed. The trap was then returned to its initial position. All of the individual beetles were preserved in cool conditions (5–8 °C) for taxonomic identification [15].

### 2.1. Effect of Lure Position in the Trap

The traps were arranged in blocks, with each block containing one trap per treatment (i.e., lure position). The number of blocks varied from three to six in each experiment. The traps within the blocks were separated by 8–10 m, and each block was sited at least 30 m apart. The traps were inspected at intervals of several days (preferably twice weekly) when captured insects were recorded and removed. At each inspection, the trap positions within a block were rotated clockwise.

The above methodology was used to test three lure positions:Low: the lure vial was inserted sealed and upside down into the bottom space (i.e., inside the narrowest part of the funnel) (Figure 1);Medium: the lure vial was inserted sealed and upside down into the middle space between the white lids (Figure 1);High: the lure vial was inserted sealed and upside down into the top space between the white lids (Figure 1).

The potential effect of lure position was assessed in fields with:(a)no vegetation for all or part of the experiment, or small, sparse vegetation, e.g., maize or soybean, when the soil was initially bare but later had a low density of small, sparse plants. In this case, there were no obstacles to sunlight reception on the ground, or to air and beetle movement. In the event of prolonged monitoring of *A. sordidus*, the soil was bare after the winter crop was harvested for silage;(b)dense vegetation, e.g., winter wheat and alfalfa, and gramineae meadows. Plant density was very high (e.g., around 500 stems per m^2^ with height ranging between 1 m and 1.5 m in winter wheat). This meant that any soil point was under shadow due to dense leaf cover and that any air/beetle movement was restricted because there were plant tissues, mainly leaves, every centimeter. Consequently, beetles could not fly horizontally any significant distance, nor could pheromone plumes move rapidly either. This ‘crop-density effect’ was lower only in the *A. ustulatus* trial, in which blocks were placed inside fields with full-grown maize, resulting in beetle and air movement in the interrow being reduced less than it was in winter wheat, alfalfa and graminae meadows.

A summary of the 25 trials is reported in Table 2.

### 2.2. Effect of Vegetation Density on Beetle Captures

We assessed the potential effect of vegetation density on the trap captures of three species (*A. sordidus*, *A. brevis* and *A. ustulatus*) in northeast Italy (45°37′26.6″ N 12°56′03.9″ E 45.62.40, 12.93.44). Fairly homogeneous plots, including several fields, were selected for the trial. The fields selected for this trial were chosen on the basis of their previous wireworm population levels and are part of a long-term wireworm monitoring program, described in Furlan [3], which took place across northeastern Italy from 1993 to 2014 [3,16]. Fields with similar wireworm densities were used when captures in the previous two years ranged from 0.1–0.5 *A. brevis* per trap, from 0.5–1.5 *A. sordidus* per trap, and from 0.7–4 *A. ustulatus* per trap.

The following comparison trials were carried out on *A. sordidus* and *A. brevis*.

#### 2.2.1. Dense Vegetation vs. Sparse Vegetation (2015–2016, 3 and 5 Replications Respectively)

Treatment 1: fields with no or sparse vegetation, i.e., maize or soybean when the soil is initially bare, and later with small and sparse plants;

Treatment 2: winter wheat fields (from BBCH 45–50 on) with no plants cut around Yf traps;

Treatment 3: winter wheat fields with a 50 cm Ø of plants cut around Yf traps.

The distance between the traps was about 100–150 m.

#### 2.2.2. Trap Position in and Outside the Target Field—*Agriotes sordidus* (2000–2002 with 3, 6 and 8 Replications Respectively)

Treatment 1: a Yf trap in the middle of the target field (field size: 1–1.5 ha, 20 m wide and 300–500 m long);

Treatment 2: just outside the target field;

Treatment 3: in another field 150–200 m away from the target field.

Regarding *A. ustulatus*, in 2009, Yf traps placed in maize fields (interrow 75 cm) were compared with those placed in soybean fields (interrow 45 cm) in mid-to-late June–July, when both were grown and gave the maximum soil cover. No plants were cut around the Yf traps. The soybean obviously covered much more soil.

### 2.3. Effect of Vegetation Density on Sex Ratio of Beetles Caught by Yf Traps

In 2002 and 2003, we assessed the potential effect of vegetation density on the sex ratio captures of *A. sordidus* and *A. brevis* by traps in cultivated fields in northeast Italy. Yf traps were placed in the middle of fields with no or small, sparse vegetation (i.e., maize when the soil was initially bare, and later with small, sparse plants) and compared with Yf traps placed in the middle of fields covered with a graminae meadow (Comunello farm, *A. brevis* trial) and with alfalfa (Greggio farm, *A. sordidus* trial). A 50 cm Ø of plants was cut around each Yf trap in the graminae and alfalfa meadows.

The traps were at least 50 m from each other. All of the beetles collected from the traps were identified by species [15] and sex with a 50× binocular dissecting microscope.

Table 3 reports location geocoordinates and seasonal period for assessing the sex ratio of adult *A. brevis* (2003) and *A. sordidus* (2002) as impacted by field vegetation.

### 2.4. Statistical Analysis

All of the trials in this publication were set up as factorial experiments to compare the number of beetles caught in Yf traps. Since the available data did not have Gaussian distributions (verified with the Shapiro–Wilk test), the following procedure was performed on all of the data available. The data reported in each table are the medians (number of beetles caught by the trap); the study data were analyzed with ANOVA after values had been transformed into ranks [17,18]. The separation of rank means was performed with the Tukey HSD test (*p* < 0.05). All data processing was performed with STATGRAPHICS 19^®^. All of the data reported in all tables are the medians of the sampled values.

## 3. Results

### 3.1. Effect of Lure Position in the Trap

The results of each individual trial are reported in Table 4. In this table, some of the median values are equal to zero because a number of inspections produced no captures; as a result, the value dividing the distribution in two (median) was zero.

It was ascertained that lure attractant capacity can vary greatly with trap position and vegetation or ground cover characteristics. When traps were placed in fields with dense vegetation, lure position played no significant role in catching any species; the only exception was when the lure was placed in the high position, as it captured slightly fewer *A. ustulatus* and *A. litigiosus* beetles than the low and medium positions, and far fewer *A. brevis* beetles. In bare fields or fields with low-density vegetation, e.g., newly emerged maize fields, *A. brevis* beetles were significantly more attracted to the trap when the lure was in the low position. The low position also resulted in slightly more catches of *A. lineatus* and *A. sputator* beetles than the other two positions. The high position attracted significantly fewer *A. obscurus* and *A. ustulatus* beetles than the other two positions. Position played no significant role in catching the other species.

### 3.2. Effect of Vegetation Density on Beetle Captures

Dense vegetation reduced Yf trap potential for catching *Agriotes sordidus*. Yf traps placed in winter wheat fields captured fewer *A. sordidus* beetles than Yf traps placed in nearby fields that had no or sparse vegetation, even when a 50 cm diameter of vegetation was cut around the trap. No reduction in the catches of *A. brevis* was observed (Table 5).

We observed that the number of catches always increased by up to twofold or more in Yf traps placed just outside the field with dense vegetation cover, and that they were comparable to the catch levels in a nearby bare or low-crop field. The number of beetles caught just outside the field covered with the crop was comparable with Yf captures recorded in the middle of fields with no or low-density vegetation, and in a nearby bare or low-crop field (Table 6).

Capture levels for *A. ustulatus* were similar in two common crops (maize and soybean) grown in the swarming period, with soybean being much denser at ground level than maize (Table 7).

### 3.3. Effect of Vegetation Density on the Sex Ratio of Beetles Caught in Yf Traps

Vegetation density also influenced beetle sex ratio, with no or very few female *A. brevis* and *A. sordidus* being found in traps placed in the middle of fields with dense vegetation. Females, however, were always found in traps placed in fields with no or small, sparse vegetation (Table 8).

## 4. Discussion

It was ascertained that lure attractant capacity can greatly vary with lure position in the trap and field conditions (Table 4). When the traps were placed in fields with dense vegetation, the lure position played no significant role in catches of most species. The only exception was when the lure was in the high position, as it captured fewer *A. ustulatus* and *A. litigiosus* beetles than the low and medium positions, and far fewer *A. brevis* beetles. The low lure position in the Yf is suitable for all species in all field conditions and is the best choice for *A. brevis*; the medium position should be considered for *A. brevis* only when vegetation is dense. In bare fields or fields with low-density vegetation, e.g., newly emerged maize fields, the *A. brevis* lure has to be placed in the low position. The low position is also better for catching *A. lineatus* beetles; the high position was not suitable for catching *A. obscurus* or *A. lineatus*.

Dense vegetation reduced Yf trap potential for catching *Agriotes sordidus* (Table 5). The range of trap attractiveness is short (<10 m) [19,20,21], but most of the *Agriotes* beetles generally have a strong attitude for flight. Dense vegetation interferes with flight, as well as with the likelihood of a beetle entering the range of attractiveness, thus probably causing an overall reduction in beetle catches. Dense vegetation, however, did not reduce the catches of *A. brevis* (Table 5)*,* which is mainly a crawling species [12], as is *A. obscurus* [22,23].

As regards the differences in captures of male and female beetles in bare fields and in fields covered with dense vegetation, where no or few females were captured, it was demonstrated that the synthetic pheromones of some species (e.g., *A. brevis* [24] and *A. sordidus* [25]) also attracted females, although usually in much smaller numbers than males.

It was also shown that the antennae of male and female *A. brevis* and *A. sordidus* beetles perceived the pheromone compound, as well as floral volatiles produced by leaves, in a similar manner [24,25]. Therefore, the negligible number of females caught by sex pheromone Yf traps in fields covered with dense vegetation was probably due to the fact that the traps were competing with volatiles emitted by the plants.

## 5. Conclusions

The experiments have shown that trap lure position has to be chosen according to target species and partially to vegetation soil cover (Table 9). Dense vegetation (e.g., winter wheat) was found to reduce the potential for catching *Agriotes sordidus*, but did not affect the potential for catching *A. brevis*. Generally speaking, when traps were placed just outside a field, or in a nearby field with bare/sparse vegetation, the maximum catching potential was maintained and reflected the local population density of the target species. Catches of *A. brevis*, which prefers crawling to flying, especially in early season, were more closely associated with target fields and were not negatively affected by high-density vegetation. Vegetation density, however, was found to influence beetle sex ratio, with no or very few female *A. brevis* and *A. sordidus* being found in traps placed in the middle of fields with dense vegetation. Females, however, were always found in traps placed in fields with bare or low-density vegetation. This finding may be exploited to ensure more careful monitoring of female numbers and the timing of egg-laying periods, as part of population management strategies.

### Brief Guidelines for Optimum Use of Yf Traps

Generally speaking, the most reliable way to assess click beetle population pressure when the target field is covered with dense vegetation is to position the trap just outside the target field, or in a nearby field, i.e., within about 200 m, with bare or low-density vegetation. Fields with no/sparse vegetation have no such restrictions; thus, the best position is inside the target field. The only exception is for *A. brevis* traps, which must always be placed in the target field. Given that the low lure position in the Yf is suitable for all species in all field conditions, the most suitable lure position should be chosen by consulting Table 9, which summarizes the results of all the experiments on the effect of lure position on different *Agriotes* species.

The low lure position is suitable for all species in all field conditions, and is the best choice for *A. brevis*; the medium position should be considered for *A. brevis* only when vegetation is dense. *A. brevis and A. lineatus* lures need to be placed in the low position when the field has no or sparse vegetation cover. The high lure position is unsuitable for *A. brevis* and *A. obscurus*, and should be considered for some species only. There are no restrictions on position for catching *A. sordidus*, i.e., any position is suitable.

Monitoring periods for each target species are described in the first part of the Materials and Methods section. *A. brevis* lures do not need to be replaced, but the other lures need to be renewed every 30 to 40 days when monitoring experiments run for more than one month [8,11].

The research presented here enables Yf traps to be exploited to their fullest potential, resulting in an easier and more affordable implementation of Integrated Pest Management (IPM) of soil pests, such as wireworms. The potential of Yf traps has also been confirmed for catching *A. obscurus* in Canada [26]. An effective ground-based trap has also been developed for *A. obscurus* and *A. lineatus* [27].

In addition, this research allows farmers to establish which areas and fields have a higher risk of wireworm damage [3], and to implement IPM in accordance with Directive 128/2009/EC.

## Figures and Tables

**Figure 1 insects-14-00542-f001:**
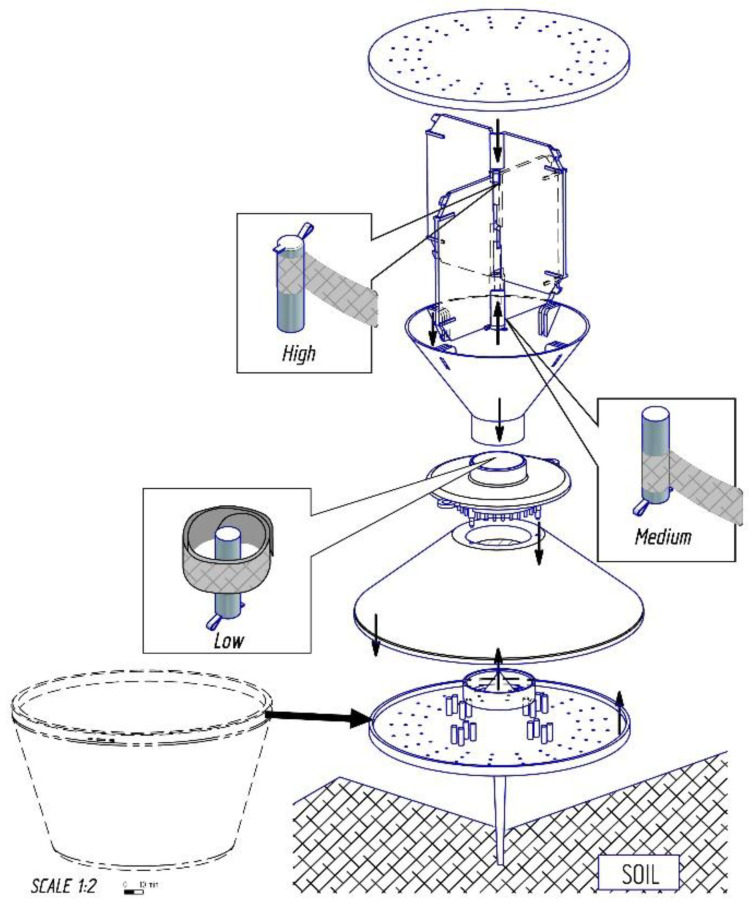
Representation of lure positions in YATLORf traps.

**Table 1 insects-14-00542-t001:** Composition of click beetle pheromone lures used in the tests.

Target Species	Active Ingredient(s)	Reference
*Agriotes brevis* Candeze	geranyl butanoate + (*E*,*E*)-farnesyl butanoate 1:1	[12]
*Agriotes lineatus* L.	geranyl butanoate + geranyl octanoate 1:1	[8]
*Agriotes litigiosus* Rossi	geranyl isovalerate	[8]
*Agriotes obscurus* L.	geranyl hexanoate + geranyl octanoate 1:1	[8]
*Agriotes proximus* Schwarz	geranyl butanoate + geranyl octanoate 1:1	[13]
*Agriotes rufipalpis* Brullé	geranyl hexanoate	[14]
*Agriotes sordidus* Illiger	geranyl hexanoate	[14]
*Agriotes sputator* L.	geranyl butanoate	[8]
*Agriotes ustulatus* Schaller	(*E*,*E*)-farnesyl acetate	[8]

**Table 2 insects-14-00542-t002:** Main characteristics of lure-position trials for catching *Agriotes* click beetles with vegetation effect assessment. S = bare soil, or soil with no-to-sparse vegetation; D = dense vegetation.

Target Species	Location	Year	Vegetation	Seasonal Period	Inspections (No.)	Replications (No.)	Coordinates
*Agriotes brevis* Candeze	Eraclea (IT)	2001	S	17 April–18 April	2	4	45°60′85″ N 12°66′76″ E
	Eraclea (IT)	2002	S	10 March–24 May	8	3	45°60′96″ N 12°66′45″ E
	Eraclea (IT)	2002	D	10 March–15 June	9	3	45°58′13″ N 12°70′23″ E
*Agriotes lineatus* L.	Lendava (SLO)	2001	S	17 May–15 June	20	4	46°33′36″ N 16°27′27″ E
	Lendava (SLO)	2002	S	15 May–17 June	23	6	46°33′36″ N 16°27′27″ E
	Ljubjana (SLO)	2002	S	15 May–3 June	23	6	46°03′31″ N 14°29′14″ E
	Ljubjana (SLO)	2001	D	15 June–6 July	14	4	46°03′31″ N 14°29′14″ E
*Agriotes litigiosus* Rossi	Eraclea (IT)	2002	S	10 June–25 July	4	3	45°60′96″ N 12°66′45″ E
	Eraclea (IT)	2002	D	12 June–27 June	5	3	45°60′99″ N 12°66′08″ E
*Agriotes obscurus* L.	Ljubjana (SLO)	2001	S	17 May–15 June	19	6	46°03′31″ N 14°29′14″ E
	Lendava (SLO)	2001	D	16 June–5 July	26	4	46°33′36″ N 16°27′27″ E
*Agriotes proximus* Schwarz	Portugal (PT)	2002	S	26 February–30 July	22	3	41°32′64″ N–8°67′41″ E
	Portugal (PT)	2003	S	18 March–5 August	18	3	41°32′64″ N–8°67′41″ E
*Agriotes sordidus* Illiger	Eraclea (IT)	2002	S	7 May–6 August	7	4	45°58′09″ N 12°70′12″ E
	Eraclea (IT)	2001	D	21 May–9 July	14	4	45°58′31″ N 12°70′69″ E
*Agriotes sputator* L.	Lendava (SLO)	2001	S	15 May–14 June	19	4	46°33′36″ N 16°27′27″ E
	Ljubjana (SLO)	2001	S	15 May–15 June	19	4	46°03′31″ N 14°29′14″ E
	Ljubjana (SLO)	2002	S	13 May–3 June	23	4	46°03′31″ N 14°29′14″ E
	Ljubjana (SLO)	2001	D	15 June–6 July	14	4	46°03′31″ N 14°29′14″ E
*Agriotes ustulatus* Schäller	Cessalto (IT)	2002	S	12 June–16 July	6	4	45°69′89″ N 12°61′46″ E
	Cessalto (IT)	2001	D	10 June–23 August	12	4	45°63′35″ N 12°67′13″ E
	Cessalto (IT)	2002	*D*	12 June–15 July	8	4	45°68′11″ N 12°57′51″ E

**Table 3 insects-14-00542-t003:** Main characteristics of the trials to assess the sex ratio of click beetles captured by Yf traps in fields with “no or small, sparse vegetation”, and fields covered with “dense vegetation”. Trials carried out in Italy in 2002.

Soil Conditions	*Agriotes brevis*	*Agriotes sordidus*
Site	Eraclea, Comunello farm	Eraclea, Greggio farm
Coordinates	45°60′96″, 12°66′45″	46°58′13″, 12°70′23″
Seasonal period	10 May–20 May	1 May–28 May
Replications (no.)	4	3
Inspections (no.)	1	4

**Table 4 insects-14-00542-t004:** Effect of lure position in Yf traps on adult click beetle catches. The values reported are the median of the samples, while the letters refer to the statistical differences using the Tukey HSD test (*p* < 0.05) on the transformed data using the rank transformation. The values in parentheses refer to the percentage respect to the maximum value of the lures for each experiment. nd stands for non-determinable. ** stands for statistical differences (*p* < 0.01), * stands for statistical differences (*p* < 0.05), ns * stands for no statistical differences (*p* > 0.05).

Target Species	Year	Site	Fields with No to Sparse Vegetation					Fields with Dense Vegetation				
			Lure Position			Date	Block	Lure Position			Date	Block
			Low	Medium	High			Low	Medium	High		
*Agriotes brevis* Candeze	2001	*Eraclea (IT)*	30 a	12 b	5.5 b	**	**					
			(100%)	(40%)	(18%)							
	2002	*Eraclea (IT)*	8 a	2 b	1.5 b	**	Ns	30.5 a	38.5 ab	4 b	**	ns
			(100%)	(25%)	(19%)			(79%)	(100%)	(10%)		
*Agriotes lineatus* L.	2001	*Lendava (SLO)*	5.5 a	3 b	2.5 b	**	*					
			(100%)	(55%)	(45%)							
	2002	*Lendava (SLO)*	0 a	0 a	0 a	*	Ns					
			(nd)	(nd)	(nd)							
	2001	*Ljubjana (SLO)*						2 a	1.5 a	2 a	*	ns
								(100%)	(75%)	(100%)		
	2002	*Ljubjana (SLO)*	8.5 a	10 a	6.5 b	**	*					
			(85%)	(100%)	(65%)							
*Agriotes litigiosus* Rossi	2001	*Cavallino (IT)*	1 a	6 a	4 a	**	Ns					
			(17%)	(100%)	(67%)							
	2002	*Eraclea (IT)*						9 a	11 a	5 b	*	**
								(82%)	(100%)	(45%)		
*Agriotes obscurus* L.	2001a	*Ljubjana (SLO)*	0 a	0 a	0 a	**	Ns					
			(nd)	(nd)	(nd)							
	2001b	*Lendava (SLO)*						0 a	0 a	0 a	*	ns
								(nd)	(nd)	(nd)		
*Agriotes proximus* Schwarz	2002	*Portugal (PT)*	24 a	17 a	23.5 a	**	Ns					
			(100%)	(71%)	(98%)							
	2003	*Portugal (PT)*	58 a	50.5 a	88 a	**	**					
			(66%)	(57%)	(100%)							
*Agriotes sordidus* Illiger	2002	*Eraclea (IT)*	9 a	16 a	16 a	**	Ns	12 a	10 a	8 a	**	ns
			(56%)	(100%)	(100%)			(100%)	(83%)	(67%)		
*Agriotes sputator* L.	2001a	*Lendava (SLO)*	0.5 a	0 a	0 a	**	Ns					
			(100%)	(0%)	(0%)							
	2001b	*Lendava (SLO)*	5.5 a	3 b	2.5 b	**	*					
			(100%)	(55%)	(45%)							
	2001	*Ljubjana (SLO)*						0 a	0 a	0 a	*	ns
								(nd)	(nd)	(nd)		
	2002	*Ljubjana (SLO)*	0 a	0 a	0 a	**	**					
			(nd)	(nd)	(nd)							
*Agriotes ustulatus* Schaller	2002a	*Cessalto (IT)*	18 ab	28 a	12.5 b	**	*	6 a	5 a	3 a	**	ns
			(64%)	(100%)	(45%)			(100%)	(83%)	(50%)		
	2002b	*Cessalto (IT)*						16.5 b	25 a	13.5 b	**	**
								(66%)	(100%)	(54%)		

**Table 5 insects-14-00542-t005:** Effect of vegetation density on median number of click beetle captures by in-field traps. No or sparse vegetation: maize or soybean (initially bare soil, later small and sparse plants); dense vegetation: winter wheat with no plants cut around the Yf trap and with plants cut around the Yf trap (50 cm Ø). The values reported are the median of the samples, and the letters refer to the statistical differences using the Tukey HSD test (*p* < 0.05) on the transformed data using the rank transformation. The comparison was made within the row.

Target Species	Years	No/Small, Sparse Vegetation		Dense Vegetation			
				No Plants Cut around Yf Trap		Plants Cut around Yf Trap (50 cm Ø)	
*Agriotes sordidus* Illiger	2000–2002	73	a	18	b		
*Agriotes sordidus* Illiger	2014–2016	450.5	a			175	b
*Agriotes brevis* Candeze	2014–2016	81	ns			56	ns

**Table 6 insects-14-00542-t006:** Effect of trap position on median number of *A. sordidus* click beetle captures in fields planted with sparse crops (maize or soybean). During the experiments, the fields were bare (before sowing or plant emergence) or had small plants in the rows (no obstacles to beetle and plume movement). “ns” stands for no statistical differences (*p* > 0.05).

Target Species	Trap Position					
	Middle of Field		Just Outside Field		In Another Field (150–200 m Away)	
*Agriotes sordidus* Illiger	364	ns	379	ns	411	ns

**Table 7 insects-14-00542-t007:** Effect of crop on the median number of *A. ustulatus* click beetle captures in maize and soybean fields. For the purpose of these experiments, the traps were placed in the middle of the crop field. “ns” stands for “no statistical differences” (*p* > 0.05).

Target Species	Crop			
	Maize		Soybean	
*Agriotes ustulatus* Schaller	1064.5	ns	1045	ns

**Table 8 insects-14-00542-t008:** Effect of crop density on *A. brevis* and *A. sordidus* males and females (median-based calculations). The values reported are the median of the samples, and the letters refer to the statistical differences using the Tukey HSD test (*p* < 0.05) on the transformed data using the rank transformation. The comparison was made within the row.

Target Species	Beetle Sex	Dense Vegetation	Bare/Sparse Vegetation
*Agriotes sordidus*	Female	0 b	8 a
	Male	150 a	23.5 b
*Agriotes brevis*	Female	0 b	2.5 a
	Male	48.5 a	29.5 a

**Table 9 insects-14-00542-t009:** Intensity of click beetle catches by traps with respective pheromone lure dispensers placed in the low, medium or high positions, summarizing all the information from the single trials. +++ high intensity, reflects maximal catches in the given trial, ++ medium intensity reflects sizeable catches but significantly lower than maximal, + low intensity reflects catches significantly lower than maximal or medium catches, n.t. not tested.

Target Species	Lure Position in Traps Placed in Fields with No or Small, Sparse Vegetation			Lure Position in Traps Placed in Fields with Dense Vegetation		
	Low	Medium	High	Low	Medium	High
*Agriotes brevis* Candeze	**+++**	**+**	**+**	**+++**	**+++**	**+**
*Agriotes lineatus* L.	**+++**	**++**	**+**	**+++**	**+++**	**+++**
*Agriotes litigiosus* Rossi	**+++**	**+++**	**+++**	**+++**	**+++**	**++**
*Agriotes obscurus* L.	**+++**	**+++**	**+**	**+++**	**+++**	**+++**
*Agriotes proximus* Schwarz	**+++**	**+++**	**+++**	**n.t.**	**n.t.**	**n.t.**
*Agriotes sordidus* Illiger	**+++**	**+++**	**+++**	**+++**	**+++**	**+++**
*Agriotes sputator* L.	**+++**	**+++**	**++**	**+++**	**+++**	**+++**
*Agriotes ustulatus* Schaller	**+++**	**+++**	**++**	**+++**	**+++**	**++**

## Data Availability

The data presented in this study are available on request from the author, stefano.bona@unipd.it.

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
