# Peer review of "The Effect of Lure Position and Vegetation on the Performance of YATLORf Traps in the Monitoring of Click Beetles (Agriotes spp., Coleoptera: Elateridae)"

_insects, 2023, doi:10.3390/insects14060542_

Round 1
Reviewer 1 Report
Dear authors,
Your research attends management requirements from the European Union for alternatives to reduce pesticide use, this is admirable. The use of English language is good and reading flows well, with minor syntax improvement recommended, if possible. Regarding there's room for improvement, please address my comments and I wish you a successful year.
Question: Although nature based is there any study suggesting dissolved pheromone in water may be somehow detrimental to the environment?
Specific recommendations
Simple summary is too similar to abstract, maybe explain why Agriotes species are detrimental, what do they affect specifically? Also, edit sentences that are identical.
Optional: join line 66 with sentence beginning in line 69.
Line 72. Remove “the” before Agriotes spp.
Line 73. Begin Objectives paragraph here, so split paragraph here.
Line 75. Important: “the Yf position in the field may have impact catch potential” is mentioned here but not listed in objectives to be determine in the study lines 79-83.
Line 93. Add comma before “and”. And check other instances when listing i.e., x, x, and is used. Instead of “many” give the number of fields (75?). Also describe the sites lat-long? elevation why chosen? Maybe a map figure to understand proximity as Slovenia and eastern Italy are close but Portugal in not.
Line 95. A photo of trap or link to permanent phot depository may be good to show here.
Line 108. Specify lure position within the trap. What is lure combination effect, this is the first mention on the text and was not on the objectives? Also, consider moving this paragraph to section 2.1 Please add block size, describe potential position effects, i.e. wind direction, others?
Line 127. As vegetation density, i.e., dense, or sparse are subjective and were not measured please include the best definition possible of the two variables in the methods section.
Line 169. Consider changing “binoculars” to “binocular dissecting microscope” for those not European.
Line 190. Change “field conditions” to “vegetation or ground cover characteristics”.
Tables and Figures
All text and Table title and legends, authors should be consistent in how the refer to the studied insects. I.e. beetles and adult click beetles should be all refer to as Agriotes spp.
Figure 1. The distance between lures should be mentioned as no scales are in the Figure.
Table 2. In headings change No/Small sparse vegetation (no.). To: No to sparse vegetation (no.)
Table 3. Why only give lure position trial of A. brevis? Top row is not a heading but a variable, remove bold font. Replications are not mentioned in the methods. Number of replications is given in Table 2 (n=3), it will be appropriate to mention replications besides the Sparse vegetation and Dense vegetation columns in Table 2. Please clarify. I believe location should be discussed early in the methods and given with a figure with coordinates to all sites there and replication # and Inspections # should be added to table 2, I believe it will make things clearer.
Table 4. Here no or small and sparse vegetation seem to represent different variables. But in Table 2 the two variables are lumped, please clarify.
Table 5. Explain what the letters a, b represent. Clarify “percentage respect to the maximum value of the lures for each experiment”. Does this represent percent of total beetles captured per Trial =experiment? So, for Agriotes brevis 2001 capture experiment in Eraclea, Italy the total captured beetles were 30 in low position, but presenting this data in percentange for the other trap captures is not recommended in lace of actual number of beetles captured.
Table 6. I don’t see the use of this table. But improving table 5 to be more informative is best.
Table 7. Doesn’t cutting plants around the trap defeats the comparison between dense and sparse vegetation? In that case the comparison is not truth no nature for those two comparisons.
The experimental unit is not specified in the methods, but I believe is the trap, so ANOVA in that same group is possible at the minimum level (2*groups) when comparing between the same block. It is difficult to infer data adequacy from table 2 as trial suggest for example 6 Trials for A. sordidus in Italy but only a number of 1 sparse and 1 dense vegetation blocks? Reported.
Table 8. This can be added as text line or joined with other data to reduce the number of tables. Spell name of beetle here and in other tables as Tables should stan alone.
Table 9. The effect of crop type is not listed in the Objectives and see comment above.
I do not comment on Discussion as I would need to see an improved methods and Results sections. I recommend you work hard at improving them.
Kind regards
With minor syntax check recommended, I think the quality of English is good and reading flows well.
Author Response
Reviewer 1
Dear authors,
Your research attends management requirements from the European Union for alternatives to reduce pesticide use, this is admirable. The use of English language is good and reading flows well, with minor syntax improvement recommended, if possible. Regarding there's room for improvement, please address my comments and I wish you a successful year.
Question: Although nature based is there any study suggesting dissolved pheromone in water may be somehow detrimental to the environment?
Specific recommendations
- Simple summary is too similar to abstract, maybe explain why Agriotes species are detrimental, what do they affect specifically? Also, edit sentences that are identical.
R: DONE
- Optional: join line 66 with sentence beginning in line 69.
- R: done
- Line 72. Remove “the” before Agriotes
- R : done
- Line 73. Begin Objectives paragraph here, so split paragraph here.
- R: done?
- Line 75. Important: “the Yf position in the field may have impact catch potential” is mentioned here but not listed in objectives to be determine in the study lines 79-83.
R: the effect of the trap position is a consequence of vegetation effect; we have modified the second goal definition: “the effect of vegetation/crop density on trap effectiveness and a trap’s most suitable field position”;
- Line 93. Add comma before “and” . And check other instances when listing i.e., x, x, and is used . Instead of “many” give the number of fields (75?). Also describe the sites lat-long? elevation why chosen? Maybe a map figure to understand proximity as Slovenia and eastern Italy are close but Portugal in not.
R: text is continuously revised by a native English, teaching English at University of Padova; No need for an Oxford comma here; as to x,X they are only needed to separate long phrases with different subjects, not used in lists;
we add a new Table 2 including all the details you ask for in the remark;
- Line 95. A photo of trap or link to permanent phot depository may be good to show here.
R: Supplied as supplementary material
- Line 108. Specify lure position within the trap. What is lure combination effect, this is the first mention on the text and was not on the objectives? Also, consider moving this paragraph to section 2.1 Please add block size, describe potential position effects, i.e. wind direction, others?
R: lure position is the treatment; high, medium, low; the sentence has been simplified to avoid any confusion and moved to Section 2.1 as suggested, thank you; as described, each block comprised 3 treatments (1. one trap with the lure in the low, or bottom position: 2. one trap with lure in the medium, or middle position, 3. one trap with the lure in the high, or top position). The distance between the traps was 8-10 m, thus the net block size was that of an equilateral triangle (10 x 10 x 10); potential position and wind effects were compensated for because the trap positions within a block were rotated at each inspection as stated.
- Line 127. As vegetation density, i.e., dense, or sparse are subjective and were not measured please include the best definition possible of the two variables in the methods section.
R: the difference between dense vegetation (meadow, winter wheat and alfalfa) and bare/sparse vegetation (bare soil or soil with few small plants is extreme and clear to any agronomist in the world; when vegetation is dense, plant density is something like 500 stems per m2, with height ranging between 1 m and 1.5 m, as in winter wheat. This means that any soil point is under shadow due to dense leaf cover and that any air/beetle movement is restricted because there are plant tissues, mainly leaves, every centimeter; consequently, beetles could not fly horizontally any significant distance and that pheromone plumes could not move rapidly either. When vegetation is bare/sparse, the opposite applies, as there are no restrictions to sunlight reception on the ground, or to air and beetle movements.
- Line 169. Consider changing “binoculars” to “binocular dissecting microscope” for those not European.
R: Done
- Line 190. Change “field conditions” to “vegetation or ground cover characteristics”.
- R: Done
- Tables and Figures
All text and Table title and legends, authors should be consistent in how the refer to the studied insects. I.e. beetles and adult click beetles should be all refer to as Agriotes spp.
R: DONE
- Figure 1. The distance between lures should be mentioned as no scales are in the Figure.
R: DONE
- Table 2. In headings change No/Small sparse vegetation (no.). To: No to sparse vegetation (no.)
R: Done
Table 3. Why only give lure position trial of A. brevis? Top row is not a heading but a variable, remove bold font. Replications are not mentioned in the methods. Number of replications is given in Table 2 (n=3), it will be appropriate to mention replications besides the Sparse vegetation and Dense vegetation columns in Table 2. Please clarify. I believe location should be discussed early in the methods and given with a figure with coordinates to all sites there and replication # and Inspections # should be added to table 2, I believe it will make things clearer.
R: Done. We have condensed the two tables into one reporting all of the site characteristics.
- Table 4. Here no or small and sparse vegetation seem to represent different variables. But in Table 2 the two variables are lumped, please clarify.
R: Now Table 3. The two variables are always lumped. No distinction was made anywhere in the text. In the table caption, the theses are now in quotation marks and in italics
- Table 5. Explain what the letters a, b represent. Clarify “percentage respect to the maximum value of the lures for each experiment”. Does this represent percent of total beetles captured per Trial =experiment? So, for Agriotes brevis 2001 capture experiment in Eraclea, Italy the total captured beetles were 30 in low position, but presenting this data in percentange for the other trap captures is not recommended in lace of actual number of beetles captured.
R: Now Table 4 The letters a and b represent the two different experiments conducted in the same year on the same site. The details are provided in Table 2
- Table 6. I don’t see the use of this table. But improving table 5 to be more informative is best.
R: now Table 5: the experiment is organized in a completely different way (see details in Section 2.2) so it would have been impossible to include the results of the experiment in the previous table. We would have had to change the table structure completely and include at least 20 empty boxes. Table 4 is already complicated and it only includes one type of experiment
- Table 7. Doesn’t cutting plants around the trap defeats the comparison between dense and sparse vegetation? In that case the comparison is not truth no nature for those two comparisons.
R: the vegetation was cut only around the trap and left very dense everywhere else so that the rest of the crop was left untouched
- The experimental unit is not specified in the methods, but I believe is the trap, so ANOVA in that same group is possible at the minimum level (2*groups) when comparing between the same block. It is difficult to infer data adequacy from table 2 as trial suggest for example 6 Trials for A. sordidus in Italy but only a number of 1 sparse and 1 dense vegetation blocks?
R: The experimental unit is the trap; the new table addresses the other remarks;
- Table 8. This can be added as text line or joined with other data to reduce the number of tables. Spell name of beetle here and in other tables as Tables should stand alone. R: we do not agree since this table reports a specific trial that is completely different from the others . Separating the different trials can help the reader to understand the effects of each experiment.
- Table 9. The effect of crop type is not listed in the Objectives and see comment above.
R: one of the objectives (b) is to discover the effect of vegetation (i.e. crops); added a specification;
Reviewer 2 Report
The authors have conducted a series of trials to improve the utilization of a pheromone-baited trap for capture of click beetle species. The objective of the work is sound and I believe their methods are equally sound, however, the presentation of their research needs considerable work before it is ready to publish. The introduction does not adequately set up the need for the work, nor the trap used. The methods do not contain enough detail for the reader to understand where these trials occurred, or when, how the vegetation density was assessed and the statistics used are not complete and do not describe all the tests conducted (based upon the tables presented in the results section). As such, given the numerous questions about the methods, the results are likewise questionable as to their validity. I am particularly concerned about the number of beetles actually captured. As presented, it would seem that for some species there were none at all captured, or very few, e.g. less than 20 in total. If this is the case, it would be better for the authors to focus on those species where they did have enough captures to properly analyse and then draw conclusions in the main text, moving those species with low or no response into a supplementary table. With the data provided, I cannot confirm that all lures were even used in these trials. In its current form it is not suitable for publication.
Below I have detailed several areas that need work. If the authors choose to improve the manuscript, I am sure additional comments/questions will follow as more details are disclosed.
L15 – can remove ‘species’ if using ‘spp.’
L18 – ‘amount of vegetation’… elaborate on whether vegetation is immediately around the trap or in the greater area, or within a certain radius around the trap
L64 – specify where this Directive is coming from… which organization?
L65 – ‘over’ rather than ‘in’
L69-71 – there is a difference between the lure and the trap which houses the lure. Please be clear that it is the pheromone lure that is proving effective. Pheromones can be deployed using a wide range of traps and the research here only looked at 1 type of trap.
L73 – this is the first mention of ‘Yf’ which needs to be explained. What is the ‘Yf’ trap? A test trap or a commercially available one? Need to provide more detail about this specific trap in the introduction.
L77 – ‘vast’? better to put a number to this… ‘vast’ is.. well… not specific
L80 – here would be a good place to define the area under consideration for this density (see L18 comment)
L81 – this is different from objective b? if so, please specify how
L82 – these experiments do not ‘span’ two decades, they occur in 2 distinct time periods: 2000-2003 and 2014-2016. To state that your experiments ‘span’ two decades is misleading.
L85 – should refer reader to Figure 1 here
Table 1 – suggest grouping the species by ‘active ingredient(s)’ rather than list the species alphabetically. The issue here will be that you have multiple species responding to the same type of lure. Simple move of A. Proximus after A. lineatus is all that’s needed.
L93 – provide the number in parentheses after each country, e.g. Italy (X), Portugal (Y) and Slovenia (Z)
L94 – better to say ‘…from 2000 to 2016 using accepted methods [8].’
L108-113 – could you provide a table detailing which traps were used in which country for which species and at what time of year and for how long each set was checked? Cannot get a sense from the methods if ALL lures were tested at the same time over the same series of weeks. I understand there’s a biology issue here in terms of the species to be captured, but did you account for that by altering which lures were out at any given period? Or just have ALL lures out?
L126-128 – need to clarify how you assessed ‘density’? percent ground cover? Did you use quadrats and assess a particular area or average several quadrats? Need more detail here. Might be better to use different descriptors, e.g. pre-crop (bare ground), post-germination (40% bare ground), mature crop (<10% bare ground)… or something like this if you didn’t actually assess the density of the vegetation.
Table 2: suggest you organize the table by Site with the species targeted in each one listed (see below for an example).
|
Site |
Target species |
Number of trials conducted |
|
|
Italy |
A. Brevis |
3 |
|
|
|
A. Litigiosus |
2 |
|
|
|
A. Sordidus |
6 |
|
Table 3 – why is this table specific to A. brevis? I think you show these details for all the sites, or put all of these in the Supplementary section. I think some of this information (seasonal period in particular) needs to be in a separate table to help the reader visualize when the trials happened and which lures were used where, when and for what purpose. One table for the lure-position trials and one table for the vegetation density.
L136 – where did these trials occur? Which country? Need a great deal more detail for this section.
L139 – ‘selected’ rather than ‘spotted’
L139-140 – which ‘long-term wireworm monitoring’ program is this? And what ‘tried-and-tested’ methodology? Need more detail on these points, particularly the location of these trials and the types of fields they are in. Again, a table with this information would be very helpful. Also, this ‘long-term wireworm monitoring’ is trapping for the larval stage, not the adult so even more reason to specify the trapping method for these rather than calling it ‘tried-and-true’.
L145 – so only lures for A. sordidus and A. brevis were deployed for these trials?
L147 – may be better to relate the vegetation to its growth stage and an estimate of percent visible bare ground
L149 and 150 – are these fields with a mature winter wheat canopy?
L151 – how many traps / species? How many replicates did you put out in each field? Or was each field a single replicate?
L152 – were these the same fields as used for B1 experiments? And why the B1 and B2 designations?
L160 – need more information about the field locations and when these trials occurred, how many traps were put in the fields.
Table 4 – details from this caption need to be in the text as well. Also you don’t have the vegetation levels listed so no need to mention that in the caption. Try this: ‘Location, geocoordinates and seasonal period for assessing the sex ratio of adult Agriotes brevis and Agriotes sordidus as impacted by field vegetation in Italy 2002.’
L180 – ‘sampled values’? you mean adult counts, yes? Better to say that. What are you ranking in this analysis? Should state that. Why use ‘medians’ as opposed to means? Not clear what it is you ranked or why. This needs to be more clearly described.
Table 5: caption should ONLY refer to what is in caption 5 and not ‘subsequent ones’. For the values in parentheses, are these referring to the percentage of the catch that was counted in the ‘low’, ‘medium’, or ‘high’ lure positions? Not clear why you analysed ‘date’ in this experiment. It would seem to me that you should be pooling the total number of beetles captured in each trap (over the entire seasonal period of the trial) and then generating a mean and standard error and compare using a standard parametric test to compare lure placement. It’s also not surprising that there was a difference across dates simply due to the nature of beetle emergence, flight and then dispersal. I don’t know what is gained by analysing the data in this manner. Similarly for block – it’s good to know that there was a difference (in some cases) but what does it matter for your experimental purposes? You were testing the location of the lure as impacted by the vegetation – I think your stats would be stronger if you pooled the captures across all dates.
L186-199 – would be helpful to have an overall summary of the actual numbers that you captured. It is concerning that you used rankings when you had actual count data (there are numerous ways to handle 0s, even removing them from the analysis is an option). It suggests that you possibly had very few captures, and are trying to hide that fact, or that you are trying to analyse too much with a limited data set. Cannot fully evaluate this without seeing at least a summary of the actual data.
Table 6 – how are you calculating ‘intensity’? this needs to be described in the methods if used here
Table 7 – are these actual counts? Or again, just the median values? Would be far better to report the mean number of beetles captured here. What test was used to determine these letters? Need to indicate that in the caption. Why ‘A’ and ‘a’ and ‘b’? How can 450.5 NOT be different from 73 and 81? This seems wrong.
L230 – ‘plum movement’? what is this?
Table 8 – if there is no significant difference, the no need to put letters – that just adds clutter to the table. Same for Table 9.
L235 – you indicate ‘ns = no statistical differences’ but then use ‘a’ in the table itself.
Table 10: This one could have been done with a chi-square test where the expected ratio is 50%. What test did you use on these percentages? Not listed in the methods.
L264 – if the discussion is going to talk about interfering with flight, then more description of their flight needs to be included in the introduction to better set up the need to study vegetation density. That also leads to the question of plant stage (versus density) particularly when you’re comparing bare ground with some time post-germination and then full canopy or a meadow (?)
L277 – unless the studies in 24 and 25 specifically used the crops that you were using in this study, this statement is total hand waving. If those studies did use the same crops, then state that outright.
Overall the English language is fine. There just wasn't enough detail in the various sections for the reader to fully understand the design.
Author Response
Find attached our replies to Reviewer 2.

Reviewer 3 Report
In general, this manuscript is well written, but there are some confusing sections that need better clarification. I uploaded an annotated version of the manuscript with my comments and suggestions, which are minor.

Author Response
Reviewer 3
Comments and Suggestions for Authors
In general, this manuscript is well written, but there are some confusing sections that need better clarification. I uploaded an annotated version of the manuscript with my comments and suggestions, which are minor.
- know you defined this in the Abstract, but you should also define this in the Introduction the first time it is used.
R: DONE
- Pag, 2, L86: if you define Yf as YATLORf in the Introduction, you can probably just us the abbreviation here. Or start the sentence with YATLORf (Yf) and use the abbreviation here.
R: DONE
- L160: Only in 2002?
R: A. sordidus in 2002, A. brevis in 2003, we have specified this;
- I think it would be better to say 'During 2000-2002 and 2014-2016', to be better synchronized with Table 7.
R: DONE
- The sparse and dense vegetation trials don't match the number of trials in column 3 for this species and the one below.
R: you are right, modified
- statistical differences between/among what variables?
R3 : the treatments, as declared are Low, Medium, High lure positions, therefore between the 3 positions;
- Pag, 6, Table: Are these to indicate differences among dates and blocks within an experiment? If you are comparing dates, shouldn't you run a repeated measures analysis of variance? In general, this table is confusing.
R: trap positions were swapped on every inspection. As inspections were not at fixed intervals, it was impossible to use a repeated measures analysis of variance. In any case, our approach was more conservative as the significance in our way was lower;
11
- Is this based on data pooled from all lure-height experiments? If so, that and the years included should be noted
R: no, each experiment was analyzed separately
- The letter designations for statistical differences? Are you comparing within rows or columns, and what is the significance of versus lowercase letters?
R: always within rows; there is no difference between uppercase and lowercase; in order to avoid any confusion, we have now used only the lowercase;
- Is this data from more than 1 experiment?
R: no, each experiment was analyzed separately
- The letter designations for statistical differences? Are you comparing within rows or columns, and what is the significance of uppercase versus lowercase letters?
R: along the row; case marks the difference between the treatments but there is no difference between uppercase and lowercase; in order to avoid any confusion, we have now used only the lowercase;
- Table 9: Is this data from 1 experiment or pooled from several? If you are pooling data that should be noted.
R: each experiment was analyzed separately
- Table 10: Results of experiment outlined in Table 4, thus from 2002?
R: yes
- Table 10: Is your letter system comparing within rows, columns, or within species and columns?
R: within the row, since the treatments are Dense and Bare/sparse vegetation.
- That appears relatively significant, as it affected 3 out of the 8 species (38%) you have listed. Thus that is a significant role.
R: yes; we have replaced “significant” with “meaningful” so that the sentence reads “When the traps were placed in fields with dense vegetation, the lure position played no MEANINGFUL role in the catches of any species. The only exception was when the lure was in the high position, as it captured fewer A. ustulatus and A. litigiosus beetles than the low and medium positions and far fewer A. brevis beetles”.
- Brief Guidelines…: What about timing of trap deployment? Shouldn't that be included in guidelines for optimum use of monitoring traps?
R: we have described the guidelines for the specific outputs of this manuscript; in any case, timing of trap deployment is obviously a crucial factor; we have added a sentence to complete the guidelines;
Round 2
Reviewer 2 Report
Authors have addressed some but not all of the comments, despite indicating this in their ‘response to reviewer’, plus there are still some areas which require clean up. It is still unclear from their data if their analysis is appropriate. The discussion and conclusion are largely a reiteration of their results with a heavy emphasis on sprinkling in references without indicating whether they agree or disagree with their obtained results. There is little comparison of the species in terms of why they could differ in their response to the lure placement and/or why vegetation would matter, or what this contributes to our understanding of click beetle behavior.
“L18 – ‘amount of vegetation’… elaborate on whether vegetation is immediately around the trap or in the greater area, or within a certain radius around the trap
R: modified to: the extent of the vegetation density all around the trap;”
On L19 in the revised version, it still says only ‘amount of vegetation’
“L80 – here would be a good place to define the area under consideration for this density (see L18 comment)
R: DONE”
On L71 in the revised version, it reads “b) the effect of vegetation/crop density on trap effectiveness and the trap’s most suitable field position;” which does not address the comment. The authors could have provided a range of plant densities, or growth stages here, but did not. Further, the addition of the latter part ‘trap’s most suitable field position’ is unclear. This objective would be better stated as “the effect of vegetation/crop density and trap location within the field on trap effectiveness, where crop density was evaluated using….” The authors remain unclear about how the vegetation was quantified.
“L126-128 – need to clarify how you assessed ‘density’? percent ground cover? Did you use quadrats and assess a particular area or average several quadrats? Need more detail here. Might be better to use different descriptors, e.g. pre-crop (bare ground), post-germination (40% bare ground), mature crop (<10% bare ground)… or something like this if you didn’t actually assess the density of the vegetation.
R: the main purpose of this work was to evaluate whether the trap Yf being tested was able to operate in a variety of conditions. There was no need to associate particular plant cover densities with the ability to capture beetles; the sole purpose was to verify whether the trap would work in different conditions, such as those shown in Table 2”
The authors attempt to address this on Lines 112-120, but in the ‘sparse’ plots the plants are growing and while the density may not increase, the shading of the soil could likely change over the time of their sampling, particularly when the sampling spans from May through August. How was this accounted for in the analysis? Had this been quantified in some manner the analysis could have included ‘percent shading’ or ‘plant height’ to capture the change over time. On L138 the authors again use the term ‘sparse’ yet both maize or soybean would be planted at a specific spacing (much like the winter wheat I would think). Was there poor germination in these fields to impact the density?
“L139-140 – which ‘long-term wireworm monitoring’ program is this? And what ‘tried-and-tested’ methodology? Need more detail on these points, particularly the location of these trials and the types of fields they are in. Again, a table with this information would be very helpful. Also, this ‘long-term wireworm monitoring’ is trapping for the larval stage, not the adult so even more reason to specify the trapping method for these rather than calling it ‘tried-and-true’.
R: Yes, we refer to “long-term wireworm monitoring” described in Furlan, 2014; some fields monitored to assess larval populations were chosen for this experiment in order to have, or at least potentially have similar beetle populations (i.e. deriving from similar larval populations); the method can be clearly understood reading M&M of the cited paper (Furlan 2014);”
This, again, does not answer the question. The reader should not have to seek out other papers to understand what you did, or why. A bit of explanation of these trials is all that is necessary here. For example ‘These fields were selected from those used as part of an ongoing long-term wireworm monitoring program, described in Furlan [3], which occurred annually across northern Italy from *year* to present [3, 16].’ As the methods to capture wireworm are quite different from those to capture adult click beetles, is there a need to specify ‘tried and true methodology’ at all? Were you using their methods? And if so how? All I’m asking is for some elaboration on what ‘tried and tested method’ are you following and how long the long-term wireworm program has been going on, and where.
Other areas of concern:
L72 in the revised version – better to say ‘c) the effect of vegetation (presence and density) on sex ratio of captured species’
L119 – the last sentence is identical to L124.
L131 – as the ‘homogeneous’ is referring to the wireworm densities, it might be better say ‘Fields selected for this trial were chosen based upon their historical wireworm population levels. These fields were selected from those used as part of an ongoing long-term wireworm monitoring program, described in Furlan [3], which occurred annually across northern Italy from *year* to present [3, 16]. Fields with similar wireworm densities were used where captures in the previous year ranged from 0.1 – 0.5 A. brevis per trap and 0.5-1.5 A. sordidus per trap. A. ustulatus populations in the previous two years….’
L137 – Table 2 needs to have the data for these sites as well. Only 2001-2003 are included in Table 2 at present. Also, I think the data from Table 3 could be included in Table 2 if a row is used to denote the experiment or objective. For example, under ‘Target species’ could put ‘(lure placement’), then add rows to include the 2015-2016 trials and these could have ‘Target species (vegetation density)’ in the row above their information with the last experiment ‘Target species (sex ratio)’ having the fields and all that information used for these experiments.
L140 – if the authors can provide growth stage for the winter wheat, can this not be provided for the maize and soybean?
L152 – should this be ‘3.3 Effect of vegetation density…’?
L172 – delete ‘the sampled values’
Table 4 – suggest moving the letters as superscripts onto the number in the previous row – this would remove excess rows from the table and make it easier to read
Tables 6 and 7 – with the data being non-significant, these can be presented in the text.
Table 8 – the caption says ‘sex ratio’ but the data are showing the values for females and males. Better to say ‘Effect of crop density on A. brevis and A. sordidus males and females (calculated based on medians)…..’ the sex ratio would be something like 4.5:1, males: female which is not what you are presenting, nor what you analysed. You compared captures of males and females.
L249 – ‘sex ratio’ – better to change this to ‘captures of males and females’
Discussion – this is very sparse. The authors could elaborate on the behavior of the click beetles, the pheromone components (particularly where females are attracted) and why this could be the case. The authors rely heavily on providing a citation but do not indicate whether the results confirm or refute their findings.
Table 9 – what does ‘n.t.’ denote – need to include that in the caption as well.
Overall the language is much improved and things are more clear.
Round 3
Reviewer 2 Report
The authors have addressed the specific responses provided during last review and provided further clarification to points that were vague or ambiguous. While there are still areas which could be improved, the authors have decided to not adjust their manuscript, e.g. Discussion. While this creates an overall weaker manuscript, it is the authors decision to adjust, or not. One final, specific, point that needs adjustment is detailed below.
L70 - Objective 3 - reads 'sex ratio' when it should read 'number of males and females'
Author Response
L70 - Objective 3 - reads 'sex ratio' when it should read 'number of males and females'
Done.